# GenArtist: Multimodal LLM as an Agent for Unified Image Generation and Editing

Zhenyu Wang[1] *    Aoxue Li[2]    Zhenguo Li[2]    Xihui Liu[3] †

[1] Tsinghua University   [2] Noah's Ark Lab, Huawei   [3] The University of Hong Kong
wangzy20@mails.tsinghua.edu.cn, lax@pku.edu.cn,
Li.Zhenguo@huawei.com, xihuiliu@eee.hku.hk

## Abstract

Despite the success achieved by existing image generation and editing methods, current models still struggle with complex problems including intricate text prompts, and the absence of verification and self-correction mechanisms makes the generated images unreliable. Meanwhile, a single model tends to specialize in particular tasks and possess the corresponding capabilities, making it inadequate for fulfilling all user requirements. We propose **GenArtist**, a *unified* image generation and editing system, coordinated by a multimodal large language model (MLLM) agent. We integrate a comprehensive range of existing models into the tool library and utilize the agent for tool selection and execution. For a complex problem, the MLLM agent decomposes it into simpler sub-problems and constructs a tree structure to systematically plan the procedure of generation, editing, and self-correction with step-by-step verification. By automatically generating missing position-related inputs and incorporating position information, the appropriate tool can be effectively employed to address each sub-problem. Experiments demonstrate that GenArtist can perform various generation and editing tasks, achieving state-of-the-art performance and surpassing existing models such as SDXL and DALL-E 3, as can be seen in Fig. 1. Project page is https://zhenyuw16.github.io/GenArtist_page/.

## 1 Introduction

With the recent advancements in diffusion models [17, 10], image generation and editing methods have rapidly progressed. Current improvements in image generation and editing can be broadly categorized into two tendencies. The first [40, 41, 37, 6, 1] involves training from scratch using more advanced model architectures [41, 36] and larger-scale datasets, thereby scaling up existing models to achieve a more general generation or editing capability. These methods can usually enhance the overall controllability and quality of image generation. The second is primarily about finetuning or additionally designing pre-trained large-scale image generation models on specific datasets to extend their capability [42, 23, 4] or enhance their performance on certain tasks [25, 18]. These methods are usually task-specific and can demonstrate advantageous results on some particular tasks.

Despite this, current image generation or editing methods are still imperfect and confront some urgent challenges on the way to building a human-desired system: 1) The demand for image generation and editing is highly diverse and variable, like various requirements for objects and backgrounds, numerous demands about various operations in text prompts or instructions. Meanwhile, different models often possess different strengths and focus. General models may be weaker than some finetuned models in certain aspects, but they can exhibit better performance in out-of-distribution data.

---

*This work is done when Zhenyu Wang was intern in Huawei
†Corresponding author

38th Conference on Neural Information Processing Systems (NeurIPS 2024).

*text-to-image generation*

An icy landscape. A vast expanse of snow-covered mountain peaks stretches endlessly. Beneath them is a dense forest and a colossal frozen lake. Three people are boating in three boats separately in the lake. Not far from the lake, a volcano threatens eruption, its rumblings felt even from afar. Above, a ferocious red dragon dominates the sky and commands the heavens, fueled by the volcano's relentless energy flow.

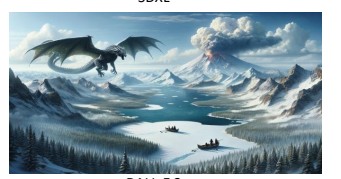

SDXL

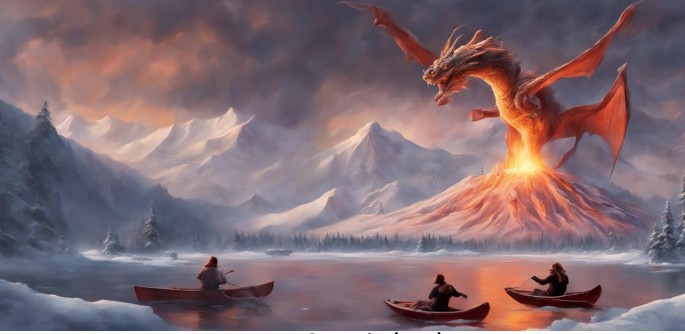

GenArtist (ours)

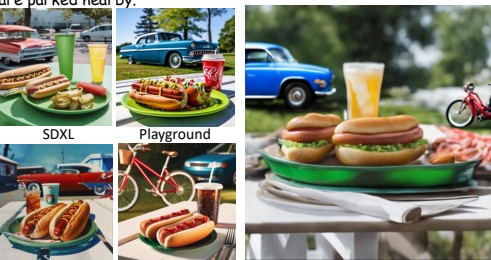

DALL-E 3

Two hot dogs sit on a green plate near a soda cup which are sitting on a white picnic table, while a red bike on the right of a blue car are parked nearby.

A restroom features black and white checkered flooring, two toilets of which has a black seat and lid and the other a white seat and lid, two black sinks.

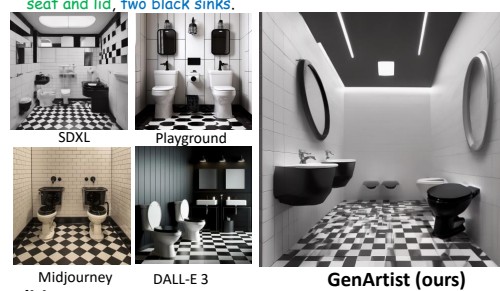

SDXL  Playground

Midjourney  DALL-E 3  **GenArtist (ours)**

SDXL  Playground

Midjourney  DALL-E 3  **GenArtist (ours)**

*image editing*

In a vibrant summer village in the daylight, the snow has melted and lush green trees dot the landscape, while houses blend seamlessly. In the distance, some mountains retain a hint of snow, a small river is in the middle of the village.

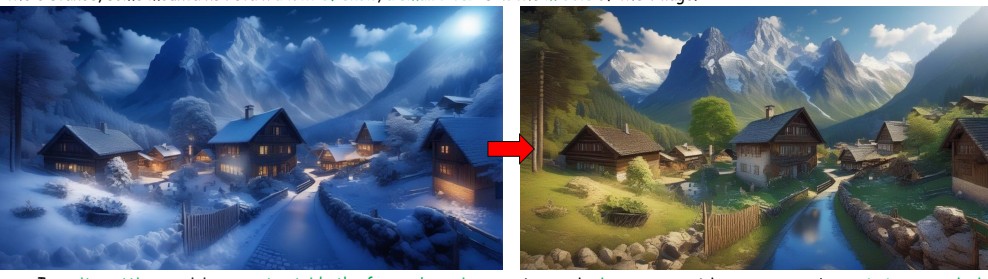

In a city setting, envision a goat outside the fence observing the cows, while removing the cow nearest to the goat.

Leave the laptop open with a mouse nearby, omitting any telephone on the table, and imagine this scene rendered in an oil painting style.

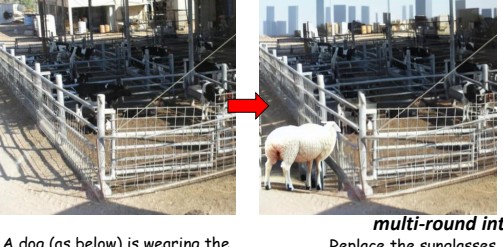

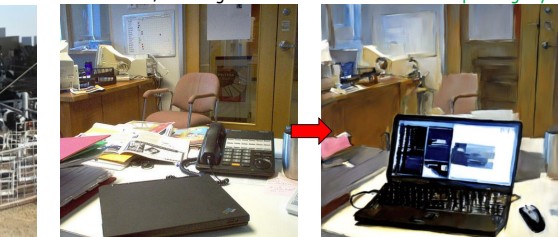

*multi-round interactive image generation*

A dog (as below) is wearing the sunglasses (as below) on the beach.

Replace the sunglasses with the below one.

Move them to a forest.

Let the cat accompany him on his right while changing to the art style.

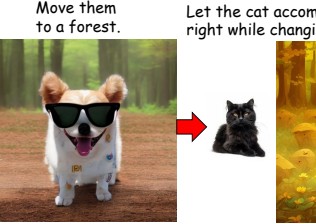

Figure 1: **Visualized examples from GenArtist.** It can accomplish various tasks, achieving unified generation and editing. For text-to-image generation, it obtains greater accuracy compared to existing models like SDXL and DALL-E 3. For image editing, it also excels in complex editing tasks.

Therefore, it is nearly impossible for a well-trained model to meet all human requirements, and the use of only a single model is often sub-optimal. 2) Models still struggle with complex problems, such as lengthy and intricate sentences in text-to-image tasks or complicated instructions with multiple steps in editing tasks. Scaling up or finetuning models can alleviate this issue. However, since texts are highly variable, flexible, and can be easy to combine, there are always complex problems that a trained model cannot effectively handle. 3) Although meticulously designed, models still inevitably encounter some failure cases. Generated images sometimes fail to accurately correspond to the content of user prompts. Existing models lack the ability to autonomously assess the correctness of generated images, not to mention self-correcting them, making generated images unreliable. What we truly desire, therefore, should be *a unified image generation and editing system*, which can satisfy nearly all human requirements while producing reliable image results.

In this paper, we propose a unified image generation and editing system called **GenArtist** to address the above challenges. Our fundamental idea is to utilize a multimodal large language model (MLLM) as an AI agent, which acts as an "artist" and "draws" images according to user instructions. Specifically, in response to user instructions, the agent will analyze the user requirements, decompose complex problems, and conduct planning comprehensively to formulate the specific solutions. Then, it executes image generation or editing operations by invoking external tools to meet the user demands. After images are obtained, it finally performs verification and correction on the generated results to further ensure the accuracy of the generated images. The core mechanisms of the agent are:

**Decomposition of intricate text prompts.** The MLLM agent first decomposes the complex problems into several simple sub-problems. For complicated text prompts in generation tasks, it extracts single-object concepts and necessary background elements. For complex instructions in editing tasks, it breaks down intricate operations into several simple single editing actions. The decomposition of complex problems significantly improves the reliability of model execution.

**Planning tree with step-by-step verification.** After decomposition, we construct a tree structure to plan the execution of sub-tasks. Each operation is a node in the tree, with subsequent operations as its child nodes, and different tools for the same action are its sibling nodes. Each node is followed by verification to ensure that its operation can be executed correctly. Then, both generation, editing, and self-correction mechanisms can be incorporated. Through this planning tree, the proceeding of the system can be considered as a traversal process and the whole system can be coordinated.

**Position-aware tool execution.** Most of object-level tools require position-related inputs, like the position of the object to be manipulated. These necessary inputs may not be provided by the user. Existing MLLMs are also position-insensitive, and cannot provide accurate positional guidance. We thus introduce a set of auxiliary tools to automatically complete these position-related inputs, and incorporate position information for the MLLM agent through detection models for tool execution.

Our main contributions can be summarized as follows:

- We propose GenArtist, a unified image generation and editing system. The MLLM agent serves as the "brain" to coordinate and manage the entire process. To the best of our knowledge, this is the first unified system that encompasses the vast majority of existing generation and editing tasks.
- Through viewing the operations as nodes and constructing the planning tree, our MLLM agent can schedule for generation and editing tasks, and automatically verify and self-correct generated images. This significantly enhances the controllability of user instructions over images.
- By incorporating position information into the integrated tool library and employing auxiliary tools for providing missing position-related inputs, the agent performs tool selection and invokes the most suitable tool, providing a unified interface for various tasks in generation and editing.

Extensive experiments demonstrate the effectiveness of our GenArtist. It achieves more than 7% improvement compared to DALL-E 3 [1] on T2I-CompBench [18], a comprehensive benchmark for open-world compositional T2I generation, and also obtains the state-of-the-art performance on the image editing benchmark MagicBrush [61]. As can be seen in the visualized examples in Fig. 1, GenArtist well serves as a unified image generation and editing system.

## 2 Related Work

**Image generation and editing.** With the development of diffusion models [10, 17], both image generation and editing have achieved remarkable success. Many general text-to-image generation [41,

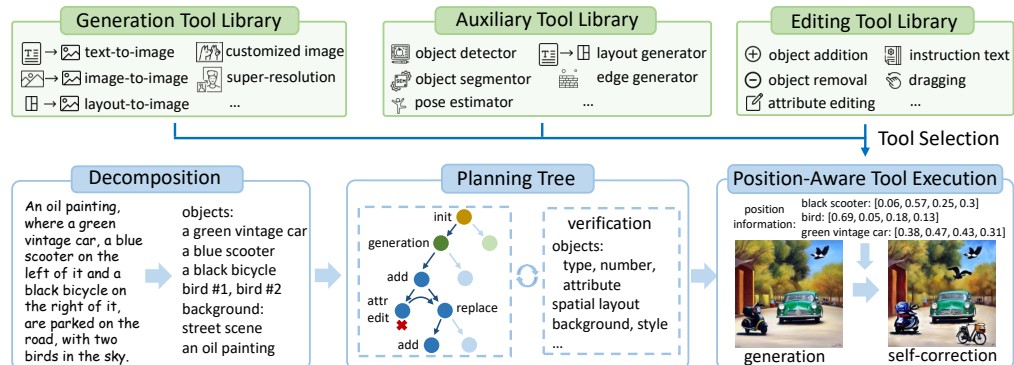

Figure 2: **The overview of our GenArtist.** The MLLM agent is responsible for decomposing problems and planning using a tree structure, then invoking tools to address the issues. Employing the agent as the "brain" effectively realizes a unified generation and editing system.

43, 37, 6] and editing methods [2, 61, 45, 15] have been proposed and achieved high-quality generated images. Based on these general models, many methods conduct finetuning or design additional modules for some specialized tasks, like customized image generation [42, 21, 23, 30], image generation with text rendering [5, 4], exemplar-based image editing [56, 8], image generation that focuses on persons [53]. Meanwhile, some methods aim to improve the controllability of texts over images. For example, ControlNet [62] controls Stable Diffusion with various conditioning inputs like Canny edges, [50] adopts sketch images for conditions, and layout-to-image methods [24, 54, 25, 7] synthesize images according to the given bounding boxes of objects. Despite the success, these methods still focus on specific tasks, thus unable to support unified image generation and editing.

**AI agent.** Large language models (LLMs), like ChatGPT, Llama [48, 49], have demonstrated impressive capability in natural language processing. The involvement of vision ability for multimodal large language models (MLLMS), like LLaVA [26], Claude, GPT-4 [34], further enables the models to process visual data. Recently, LLMs begin to be adopted as agents for executing complex tasks. These works [57, 44, 29] apply LLMs to learn to use tools for tasks like visual interaction, speech processing, compositional visual tasks [16], software development [38], gaming [11], APP use [60] or math [55]. Recently, the idea of AI agents has also begun to be applied to image generation related tasks. For example, [25, 13] design scene layout with LLMs, [52] utilizes LLMs to assist self-correcting, [51, 59] target at MLLMs in complex text-to-image generation problems, and [39] leverages LLM for model selection in the text-to-image generation task.

## 3 Method

The overview of our GenArtist is illustrated in Fig. 2. The MLLM agent coordinates the whole system. Its primary responsibilities center around decomposing the complicated tasks and constructing the planning tree with step-by-step verification for image generation, editing, and self-correction. It invokes tools from an image generation tool library and an editing tool library to execute the specific operations, and an auxiliary tool library serves to provide missing position-related values.

### 3.1 Planning Tree with Step-by-Step Verification

**Decomposition.** When it comes to complicated prompt inputs, existing methods usually cannot understand all requirements, which hurts the controllability and reliability of model results. The MLLM agent thus first decomposes the complex problems. For generation tasks, it decomposes both object and background information according to the text prompts. It extracts the discrete objects embedded within the text prompts, along with their associated attributes. For background information, it mainly analyzes the overall scene and image style required by the input texts. For editing tasks, It decomposes complex editing operations into several specific actions, such as add, move, remove, into simple editing instructions. After decomposition, the simpler operations can be relatively easier to address, which thus improves the reliability of model execution.

**Tree construction.** After decomposition, we organize all operations into a structure of tree for planning. Such a tree primarily consists of three types of nodes: initial nodes, generation nodes, and editing nodes. The initial node serves as the root of the tree, marking the beginning of the system. Generation nodes are about image generation using tools from the generation tool library, while editing nodes are about performing a single editing operation using the corresponding tools from the editing tool library. For pure image editing tasks, the generation nodes will be absent.

In practice, as the correctness of generated images cannot be guaranteed, we introduce the self-correction mechanism to assess and rectify the results of generation. Each generation node thus has a sub-tree consisting entirely of editing nodes for self-correction. After the tools in the generation nodes are invoked and verification is conducted, this sub-tree will be adaptively generated by the MLLM agent. Specifically, after verification, we instruct the MLLM agent to devise a series of corresponding editing operations to correct the images. Take the example in Fig. 2 for example, editing actions including `"add a black bicycle"`, `"edit the color of the scooter to blue"`, `"add a bird"` should be conducted. These operations are organized into a tree structure to be the sub-tree of the generation node, allowing for specific planning of self-correction.

Each generation or editing action corresponds to a node in the tree, with its subsequent operations as its child nodes. This construction initially forms a "chain", enabling a planning chain. Then, we note that we can usually utilize different tools to address the same problem. For example, for adding an object into the image, we can employ a tool specifically designed for object addition or instruction-based editing models by translating the adding operation into text instructions. Similarly, for attribute editing, we can use attribute editing models or utilize replacement or instruction-based editing models. Moreover, numerous generation tools can achieve text-to-image generation, and varying the random seeds can also produce different outputs. We consider these nodes as siblings, all serving as child nodes of their parent nodes. They also share the same sub-tree, containing subsequent editing operations. The tool selected by the MLLM agent will be placed as the optimal child node and positioned on the far left. In this way, we establish the structure of the tree. An illustration example for the Fig. 2 case is provided in Fig. 3 (we omit some sub-trees with identical structures or adaptively generated after generation nodes, and some nodes about varying random seeds for simplicity).

Figure 3: **Illustration of the tree for planning.** The sub-tree of the "alternative generation tool" node will be adaptively generated after verification, and the sub-tree of the "instruction" node is the same as the left.

**Planning.** Once the tree is established, planning for self-correction or the whole system can be viewed as the pre-order traversal of the structure. For a particular node, its corresponding tool is invoked to conduct the operation, followed by verification to determine whether the editing is successful. If successful, the process proceeds to its leftmost child node for subsequent operations, and its sibling nodes are deleted. If unsuccessful, the process backtracks to its sibling nodes, and its sub-tree is removed. This process continues until the generated image is correct, *i.e.*, when a node at the lowest level successfully executes. We can also limit the branching factor or the number of nodes of the tree for early termination, and require the agent to return the most accurate image.

**Verification.** As described above, the verification mechanism plays a crucial role both in tree construction and the execution process. Through the multimodal perception capability of the MLLM, the agent verifies the correctness of the generated images. The main aspect of verification involves the objects contained in the text, together with their own attributes like their color, shape, texture, the positions of the objects, the relationship among these different objects. Besides, the background, scene, overall style and the aesthetic quality of generated images are also considered. Since the perception ability of existing models tends to be superior to the generative ability, employing such verification allows for effectively assessing the correctness of generated images.

It is also worth mentioning that during verification, in addition to the accuracy of the generated images, the agent is also required to assess their aesthetic quality. If the overall quality is poor, the agent will utilize different generation tools or choose different random seeds to regenerate the images,

Table 1: **GenArtist utilized tool library, including the tool names and their skills.** The main tools are from the generation tool library and the editing tool library. The following models represent all the tools used in our current version, while new models can be seamlessly added.

| Generation Tools | | Editing Tools | |
|---|---|---|---|
| skill | tool | skill | tool |
| text-to-image | SDXL [37] | | |
| text-to-image | PixArt-$\alpha$ [6] | object addition | AnyDoor [8] |
| image-to-image | Stable Diffusion v2 [41] | object removal | LaMa [47] |
| layout-to-image | LMD [25] | object replacement | AnyDoor [8] |
| layout-to-image | BoxDiff [54] | attribute editing | DiffEdit [9] |
| single-object customization | BLIP-Diffusion [23] | instruction-based | MagicBrush [61] |
| multi-object customization | $\lambda$-ECLIPSE [35] | dragging (detail) | DragDiffusion [46] |
| super-resolution | SDXL [37] | dragging (object) | DragonDiffusion [33] |
| image with texts | TextDiffuser [4] | style transfer | InST [64] |
| {canny, depth ...}-to-image | ControlNet [62] | | |

in order to ensure their overall quality. Meanwhile, as an agent-centered system, the framework is also flexible in terms of human-computer interaction. During verification, human feedback can be appropriately integrated. By incorporating human evaluation and feedback on the overall quality of the images, the quality of the generated images can be further improved.

## 3.2 Tool Library

After constructing the planning tree, the agent proceeds to execute each node by calling external tools, ultimately solving the problem. We first introduce the tools used in GenArtist. The primary tools that the MLLM agent utilizes can be generally divided into the image generation tool library and the editing tool library. The specific tools we utilize currently are listed in Tab. 1, and some new tools can be seamlessly added, allowing for the expansion of the tool library. To assist the subsequent tool selection, we need to convey information to the MLLM agent about the specific task performed by the tool, its required inputs, and its characteristics and advantages. The prompts for introducing tools consist of the following parts specifically:

- The tool skill and name. It briefly describes the tool-related task and its name, as listed in Tab. 1, such as (`text-to-image, SDXL`), (`canny-to-image, ControlNet`), (`object removal, LaMa`). It serves as a unique identifier, enabling the agent to differentiate the utilized tools.
- The tool required inputs. It pertains to the specific inputs required for the execution of the tool. For example, text-to-image models require `"text"` as input for generation, customization models also need `"subject images"` for personalized generation. Most of object-level editing tools demand instructions about `"object name"` and `"object position"`.
- The tool characteristic and advantage. It primarily provides a more detailed introduction of the tool, including its specific characteristics, serving as a key reference for the agent during tool selection. For example, SDXL can be a `general text-to-image generation model`, LMD usually controls scene layout strictly and is suitable for `compositional text-to-image generation`, where text prompts usually contain multiple objects, BoxDiff controls scene layout relatively loosely, TextDiffuser is specially designed for `image generation with text rendering`.

## 3.3 Position-Aware Tool Execution

With tool libraries, the MLLM agent will further perform tool selection and execution to utilize the suitable tool for fulfilling the image generation or editing task. Before tool execution, we compensate for the deficiency of position information in user inputs and the MLLM agent through two designs:

**Position-related input compensation.** In practice, it is common to encounter scenes where the agent selects a suitable tool but some necessary user inputs are missing. These user inputs are mostly related to positions. For example, for some complex text prompts where multiple objects exist, the layout-to-image tool can be suitable. However, users may not necessarily provide the scene layouts and usually only text prompts are provided. In such cases, due to the absence of some necessary inputs, these suitable tools cannot be directly invoked. We therefore introduce the auxiliary tool library to provide these position-related missing inputs. This auxiliary tool library mainly contains: 1) localization models like object detection [28] or segmentation [20] models, to provide position

Table 2: **Quantitative Comparison on T2I-CompBench with existing text-to-image generation models and compositional methods**. Our method demonstrates superior compositional generation ability in both attribute binding, object relationships, and complex compositions. We use the officially updated code for evaluation, which updates the noun phrase number. Consequently, some metric values for certain methods may be lower than those reported in their original papers.

| Model | Attribute Binding | | | Object Relationship | | Complex↑ |
|---|---|---|---|---|---|---|
| | Color ↑ | Shape↑ | Texture↑ | Spatial↑ | Non-Spatial↑ | |
| Stable Diffusion v1.4 [41] | 0.3765 | 0.3576 | 0.4156 | 0.1246 | 0.3079 | 0.3080 |
| Stable Diffusion v2 [41] | 0.5065 | 0.4221 | 0.4922 | 0.1342 | 0.3096 | 0.3386 |
| DALL-E 2 [40] | 0.5750 | 0.5464 | 0.6374 | 0.1283 | 0.3043 | 0.3696 |
| Composable Diffusion [27] | 0.4063 | 0.3299 | 0.3645 | 0.0800 | 0.2980 | 0.2898 |
| StructureDiffusion [12] | 0.4990 | 0.4218 | 0.4900 | 0.1386 | 0.3111 | 0.3355 |
| Attn-Exct [3] | 0.6400 | 0.4517 | 0.5963 | 0.1455 | 0.3109 | 0.3401 |
| GORS [18] | 0.6603 | 0.4785 | 0.6287 | 0.1815 | 0.3193 | 0.3328 |
| SDXL [37] | 0.5879 | 0.4687 | 0.5299 | 0.2133 | 0.3119 | 0.3237 |
| PixArt-$\alpha$ [6] | 0.6690 | 0.4927 | 0.6477 | 0.2064 | 0.3197 | 0.3433 |
| CompAgent [51] | 0.7760 | 0.6105 | 0.7008 | 0.4837 | 0.3212 | 0.3972 |
| DALL-E 3 [1] | 0.7785 | 0.6205 | 0.7036 | 0.2865 | 0.3003 | 0.3773 |
| **GenArtist (ours)** | **0.8482** | **0.6948** | **0.7709** | **0.5437** | **0.3346** | **0.4499** |

information of objects for some object-level editing tools; 2) the preprocessors of ControlNet [62] like the pose estimator, canny edge map extractor, depth map extractor; 3) some LLM-implemented tools, like the scene layout generator [25, 13]. The MLLM agent can invoke these auxiliary tools automatically if necessary, to guarantee that the most suitable tool to address the user instruction can be utilized, rather than solely relying on user-provided inputs to select tools.

**Position information introduction.** Existing MLLMs primarily focus on text comprehension and holistic image perception, with relatively limited attention to precise position information within images. MLLMs can easily determine whether objects exist in the image, but sometimes struggle with discerning spatial relationships between objects, such as whether a specific object is to the left or right of another. It is also more challenging for these MLLMs to provide accurate guidance for tools that require position-related inputs, such as object-level editing tools. To address this, we employ an object detector on the input images, and include the detected objects along with their bounding boxes as part of the prompt, to provide a spatial reference for the MLLM agent. In this way, the agent can effectively determine the positions within the image where certain tools should operate.

The prompts for the agent to conduct tool selection thus mainly consist of the following parts:

- Task instruction. Its main purpose is to clarify the task of the agent, *i.e.*, tool selection within a unified generation and editing system. Simultaneously, it takes user instructions as input and specifies the output format. We request the agent to output in the format of {"tool_name":tools, "input":inputs} and annotate missing inputs with the pre-defined specified identifier.
- Tool introductions. We input the description of each tool into the agent in the format as described earlier. The detailed information about the tools will serve as the crucial references for the tool selection process. We also state that the primary criterion for tool selection is the suitability of the tool, rather than the content of given inputs, since missing inputs can be generated automatically.
- Position information. The outputs from the object detector are utilized and provided to the MLLM agent to compensate for the lack of position information.

In summary, the basic steps for tool execution are as follows: First, determine whether the task pertains to image generation or editing. Next, conduct tool selection according to the instructions and the characteristics of the tools, and output in the required format. Finally, for missing inputs which are necessary for the selected tools, utilize auxiliary tools to complete them. Upon completing these steps, the agent will be able to correctly execute the appropriate tools, thereby initially meeting the requirements of users. The integration, selection, and execution of diverse tools significantly facilitate the development of a unified image generation and editing system.

## 4 Experiments

In this section, we demonstrate the effectiveness of our GenArtist and its unified ability through extensive experiments in image generation and editing. For image generation, we mainly conduct

Table 3: **Quantitative Comparison on MagicBrush with existing image editing methods**. Multi-turn setting evaluates images that iteratively edited on the previous source images in edit sessions.

| Settinigs | Methods | L1↓ | L2↓ | CLIP-I↑ | DINO↑ | CLIP-T↑ |
|---|---|---|---|---|---|---|
| Single-turn | Null Text Inversion [32] | 0.0749 | 0.0197 | 0.8827 | 0.8206 | 0.2737 |
| | HIVE [63] | 0.1092 | 0.0341 | 0.8519 | 0.7500 | 0.2752 |
| | InstructPix2Pix [2] | 0.1122 | 0.0371 | 0.8524 | 0.7428 | 0.2764 |
| | MagicBrush [61] | 0.0625 | 0.0203 | 0.9332 | 0.8987 | 0.2781 |
| | SmartEdit [19] | 0.0810 | - | 0.9140 | 0.8150 | 0.3050 |
| | **GenArtist (ours)** | **0.0536** | **0.0147** | **0.9403** | **0.9131** | **0.3129** |
| Multi-turn | Null Text Inversion [32] | 0.1057 | 0.0335 | 0.8468 | 0.7529 | 0.2710 |
| | HIVE [63] | 0.1521 | 0.0557 | 0.8004 | 0.6463 | 0.2673 |
| | InstructPix2Pix [2] | 0.1584 | 0.0598 | 0.7924 | 0.6177 | 0.2726 |
| | MagicBrush [61] | 0.0964 | 0.0353 | 0.8924 | 0.8273 | 0.2754 |
| | **GenArtist (ours)** | **0.0858** | **0.0298** | **0.9071** | **0.8492** | **0.3067** |

quantitative comparisons on the recent T2I-CompBench benchmark [18]. It is mainly about image generation with complex text prompts, involving multiple objects together with their own attributes or relationships. For image editing, we mainly conduct comparisons on the MagicBrush benchmark [61], which involves multiple types of text instructions, both single-turn and multi-turn dialogs for image editing. We choose GPT-4V [34] as our MLLM agent. In quantitative comparative experiments, we constrain the editing tree to be a binary tree.

## 4.1 Comparison with Image Generation Methods

We list the quantitative metric results of our GenArtist in Tab. 2 and compare with existing state-of-the-art text-to-image synthesis methods. It can be seen that our GenArtist consistently achieves better performance on all sub-categories. This demonstrates that for the text-to-image generation task, our system effectively achieves better control over text-to-image correspondence and higher accuracy in generated images, especially in the case of complicated text prompts. It can be observed that based on Stable Diffusion, both scaling-up models such as SDXL, PixArt-$\alpha$, and those methods specifically designed for this context like Attn-Exct, GORS, can achieve higher accuracy. In contrast, our approach, by integrating various models as tools, effectively harnesses the strengths of these two categories of methods. Additionally, the self-correction mechanism further ensures the accuracy of the generated images. Compared to the current state-of-the-art model DALL-E 3, our method achieves nearly a 7% improvement in attribute binding, and a more than 20% improvement in spatial relationships, partly due to the inclusion of position-sensitive tools and the input of position information during tool selection. Compared to CompAgent, a method that also employs an AI agent for compositional text-to-image generation, GenArtist achieves a 6% improvement on average, partially because our system encompasses a more comprehensive framework for both generation and self-correction. The capability in image generation of our unified system can thus be demonstrated.

## 4.2 Comparison with Image Editing Methods

We then list the comparative quantitative comparisons on the image editing benchmark MagicBrush in Tab. 3. Our GenArtist also achieves superior editing results, no matter in the single-turn or multi-turn setting, compared to both previous global description-guided methods like Null Text Inversion and instruction-guided methods like InstrctPix2Pix and MagicBrush. The main reason is that editing operations are highly diverse, and it's challenging for a single model to achieve excellent performance across all these diverse editing operations. In contrast, our method can leverage the strengths of different models comprehensively. Additionally, the planning tree can effectively consider scenarios where model execution fails, making editing results more reliable and accurate. The capability in image editing of our unified system can thus be demonstrated.

## 4.3 Ablation Study

We finally conduct the ablation study on the T2I-CompBench benchmark and list the results in Tab. 4. We present the results of Stable Diffusion as a reference. The top section includes various tools relevant to the task, including text-to-image, layout-to-image, and customized generation methods. It can be observed that through scaling up or additional design, these tools have generally achieved

Table 4: **Ablation Study on T2I-CompBench**. The upper section is about relevant tools from the generation tool library, then we study the tool selection and planning mechanisms respectively.

| Method | Attribute Binding | | | Object Relationship | | Complex↑ |
|---|---|---|---|---|---|---|
| | Color ↑ | Shape↑ | Texture↑ | Spatial↑ | Non-Spatial↑ | |
| Stable Diffusion v2 [41] | 0.5065 | 0.4221 | 0.4922 | 0.1342 | 0.3096 | 0.3386 |
| LMD [25] | 0.5736 | 0.5334 | 0.5227 | 0.2704 | 0.3073 | 0.3083 |
| BoxDiff [54] | 0.6374 | 0.4869 | 0.6100 | 0.2625 | 0.3158 | 0.3457 |
| $\lambda$-ECLIPSE [35] | 0.4581 | 0.4420 | 0.5084 | 0.1285 | 0.2922 | 0.3131 |
| SDXL [37] | 0.5879 | 0.4687 | 0.5299 | 0.2133 | 0.3119 | 0.3237 |
| PixArt-$\alpha$ [6] | 0.6690 | 0.4927 | 0.6477 | 0.2064 | 0.3197 | 0.3433 |
| + tool selection | 0.7028 | 0.5764 | 0.6883 | 0.4305 | 0.3187 | 0.3739 |
| + planning chain | 0.7509 | 0.6045 | 0.7192 | 0.4787 | 0.3216 | 0.4095 |
| + planning tree | **0.8482** | **0.6948** | **0.7709** | **0.5437** | **0.3346** | **0.4499** |

Figure 4: **Visualization of the planning tree for image generation tasks.**

better results than Stable Diffusion. After tool selection by the MLLM agent, the quantitative metrics outperform all these tools. This demonstrates that the agent can effectively choose appropriate tools based on the content of text prompts, thus achieving superior performance compared to all these tools. If we use a chain structure for planning to further correct the images, we achieve an average improvement of 3%, demonstrating the necessity of verification and correction of erroneous results. Furthermore, by utilizing a tree structure, we can further consider and handle cases where the editing tool fails, resulting in even more reliable output results. Such an ablation study illustrates the necessity of integrating multiple models as tools and utilizing tree structure for planning. The reasonableness of our agent-centric system designs can also be demonstrated.

Table 5: **Ablation Study on the position-aware tool execution on T2I-CompBench.**

| | Spatial↑ | Complex ↑ |
|---|---|---|
| w/o position information | 0.4577 | 0.4083 |
| w/ position information | 0.5437 | 0.4499 |

Regarding position-aware tool execution, we list the corresponding ablation study in Tab. 5. We evaluate the performance on the spatial and complex aspects of T2I-CompBench, as these two aspects mainly involve position-sensitive text prompts for image generation. As multi-modal large models are usually not sensitive to position information, the performance is limited without the inclusion of position information, only a slight improvement over the tool selection results. After introducing position information, which enhances spatial awareness, there is a significant improvement in both the spatial and complex aspects. This validates the reasonability of our design.

We further list some visualized generation examples in Fig. 4 to illustrate our planning tree and how the system proceeds. In the first example, as the text prompts contain multiple objects, the agent chooses the LMD tool for generation. However, there are still some errors in the image. The agent first attempts to use the attribute editing tool to change the leftmost sheep to white, but it fails. The agent further attempts to modify the color using the replace tool, but after replacement, the size of the sheep becomes too small and not very noticeable. The agent then chooses to remove the black sheep and then adds a white sheep, successfully achieving the same effect as editing color. Finally, the agent uses the object addition tool to add a goat on the right side, ensuring that the image accurately matches the text prompt in the end. In the second example, due to the lack of clarity of the hair in the BoxDiff generated image, the editing tools cannot edit so that the hair correctly matches the description of "long black hair". Therefore, the agent invokes another generation tool to guarantee the final image is correct. Some image editing examples are also provided in Fig. 5.

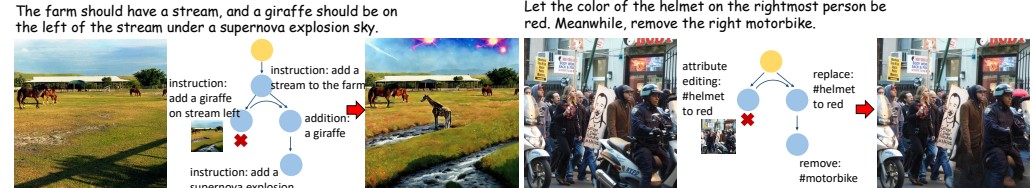

Figure 5: **Visualization of the planning tree for image editing tasks.**

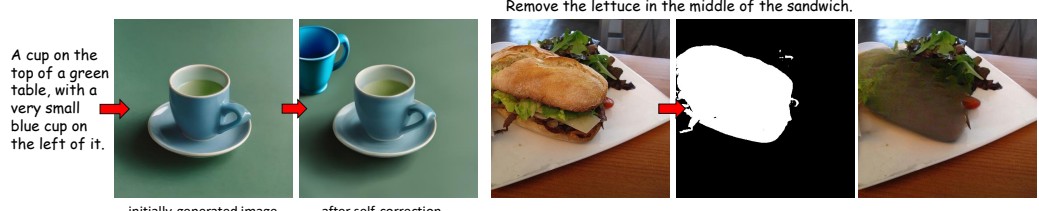

Figure 6: **The error cases of GenArtist** caused by the ability of editing tools (the left) or the wrong output of localization tools (the right).

## 4.4 Error Case Analysis

We further analyze some error cases from our GenArtist in Fig. 6. As can be seen, sometimes, despite the agent correctly planning the specific execution of tools, the limitations of the tools themselves prevent correct execution, leading to incorrect results. For example, in the first case, it is required to add a very small blue cup. However, due to the lack of fine resolution ability in existing editing tools, the generated blue cup's size is inaccurate. In addition, as shown in the second case, errors in the output of localization tools can also affect the final result. For instance, when asked to remove the lettuce in the middle of a sandwich, the segmentation model fails to accurately identify the part of the object, leading to the erroneous removal operation. Utilizing more powerful tools or incorporating some human feedback during the verification stage can effectively address this issue.

## 5 Conclusion

In this paper, we propose GenArtist, a unified image generation and editing system coordinated by a MLLM agent. By decomposing input problems, employing the tree structure for planning and invoking external tools for execution, the MLLM agent acts as the "brain" to generate high-fidelity and accurate images for various tasks. Extensive experiments demonstrate that GenArtist well addresses complex problems in image generation and editing, and achieves state-of-the-art performance compared to existing methods. Its ability in a wide range of generation tasks also validates its unified capacity. We believe our approach of leveraging the agent to achieve a unified image generation and editing system with enhanced controllability can provide valuable insights for future research, and we consider it an important step toward the future of autonomous agents.

## Acknowledgement

We gratefully acknowledge the support of Mindspore, CANN(Compute Architecture for Neural Networks) and Ascend AI Processor used for this research.

**Limitation and Potential Negative Social Impacts.** Our method employs an MLLM agent as the core for the entire system operations. Therefore, the method effectiveness depends on the performance of the MLLM used. Current MLLMs, such as GPT-4, are capable of meeting most basic requirements. For tasks that exceed the capability of GPT-4, our method may fail. Additionally, the misuse of image generation or editing could potentially lead to negative social impacts.

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

# Appendix

In the appendix, we primarily include more quantitative comparisons, along with additional visual results, to more comprehensively compare with existing state-of-the-art methods.

## A    More Quantitative Experiments

Table 6: **Quantitative Comparison on T2I-CompBench with existing text-to-image generation models and compositional methods**. The metric here is from the officially old-version code.

| Model | Attribute Binding | | | Object Relationship | | Complex↑ |
|---|---|---|---|---|---|---|
| | Color ↑ | Shape↑ | Texture↑ | Spatial↑ | Non-Spatial↑ | |
| SDXL [37] | 0.6369 | 0.5408 | 0.5637 | 0.2032 | 0.3110 | 0.4091 |
| PixArt-$\alpha$ [6] | 0.6886 | 0.5582 | 0.7044 | 0.2082 | 0.3179 | 0.4117 |
| ConPreDiff [58] | 0.7019 | 0.5637 | 0.7021 | 0.2362 | 0.3195 | 0.4184 |
| DALL-E 3 [1] | 0.8110 | 0.6750 | 0.8070 | - | - | - |
| CompAgent [51] | 0.8488 | 0.7233 | 0.7916 | 0.4837 | 0.3212 | 0.4863 |
| RPG [59] | 0.8335 | 0.6801 | 0.8129 | 0.4547 | 0.3462 | 0.5408 |
| **GenArtist (ours)** | **0.8837** | **0.8014** | **0.8771** | **0.5437** | 0.3346 | **0.5514** |

Considering that many existing methods on T2I-CompBench report results based on the official old version evaluation code, here we utilize the same old version evaluation method and list the results in Tab. 6. It can be observed that the performance improvement keeps consistently under this metric. Compared to the current state-of-the-art text-to-image method, DALL-E 3, our approach achieves over a 7% improvement in attribute binding. For shape-related attributes, the improvement is even up to 12.64%. Additionally, compared to RPG, which also utilizes MLLM to assist image generation, our method demonstrates an over 5% enhancement. This is because our GenArtist incorporates MLLM for step-by-step verification and the corresponding planning, thereby better ensuring the correctness of the images. This quantitative comparison more comprehensively demonstrates the effectiveness of our method.

## B    More Qualitative Experiments

In this section, we provide more visual analyses to further illustrate our GenArtist and to compare it more thoroughly with existing methods.

**Comparative visualized results on image generation.** We first present visual comparisons with existing methods in Fig. 7. It can be observed that our GenArtist achieves superior results in multiple aspects: 1) *Attribute Binding*. For example, in the fourth example, there are strict requirements for the clothes and pants each person is wearing. Such numerous requirements are challenging for existing methods to meet. In this case, GenArtist can continuously verify and edit to ensure all these requirements are correctly satisfied. 2) *Numeric Accuracy*. In the second example, detailed quantity requirements are given for various objects. Our method can gradually achieve the correct quantities through addition and removal operations. In contrast, even though methods like LMD+ can meet numeric accuracy, they struggle to maintain the accuracy of other aspects, such as the atmosphere of the image. 3) *Position Accuracy*. By position-aware tool execution, better position-related accuracy can be guaranteed. In the first example, although DALL-E 3 can correctly predict many other aspects, it fails to accurately place the book on the left floor, which our method can achieve. 4) *Complex Relationships*, like the complex requirements for the relationship between the panda and bamboo in the fifth example. 5) *Other Diverse Requirements*. By integrating various tools, GenArtist effectively leverages the strengths of different tools to meet diverse requirements, the ability that a single model lacks. For instance, the text requirements in the third example are better handled by our method. Such visualized results strongly demonstrate the effectiveness of our method in image generation.

**Comparative visualized results on image editing.** We further present comparisons with existing image editing methods in Fig. 8. GenArtist shows superior performance in several aspects: 1) *Highly Specific Editing Instructions*. For instance, in the first example, only a particular pizza needs to be modified, while the second example requires changes to the color and placement of a vase. Existing methods often struggle to satisfy such specific requirements. 2) *Reasoning-based Instructions*. The

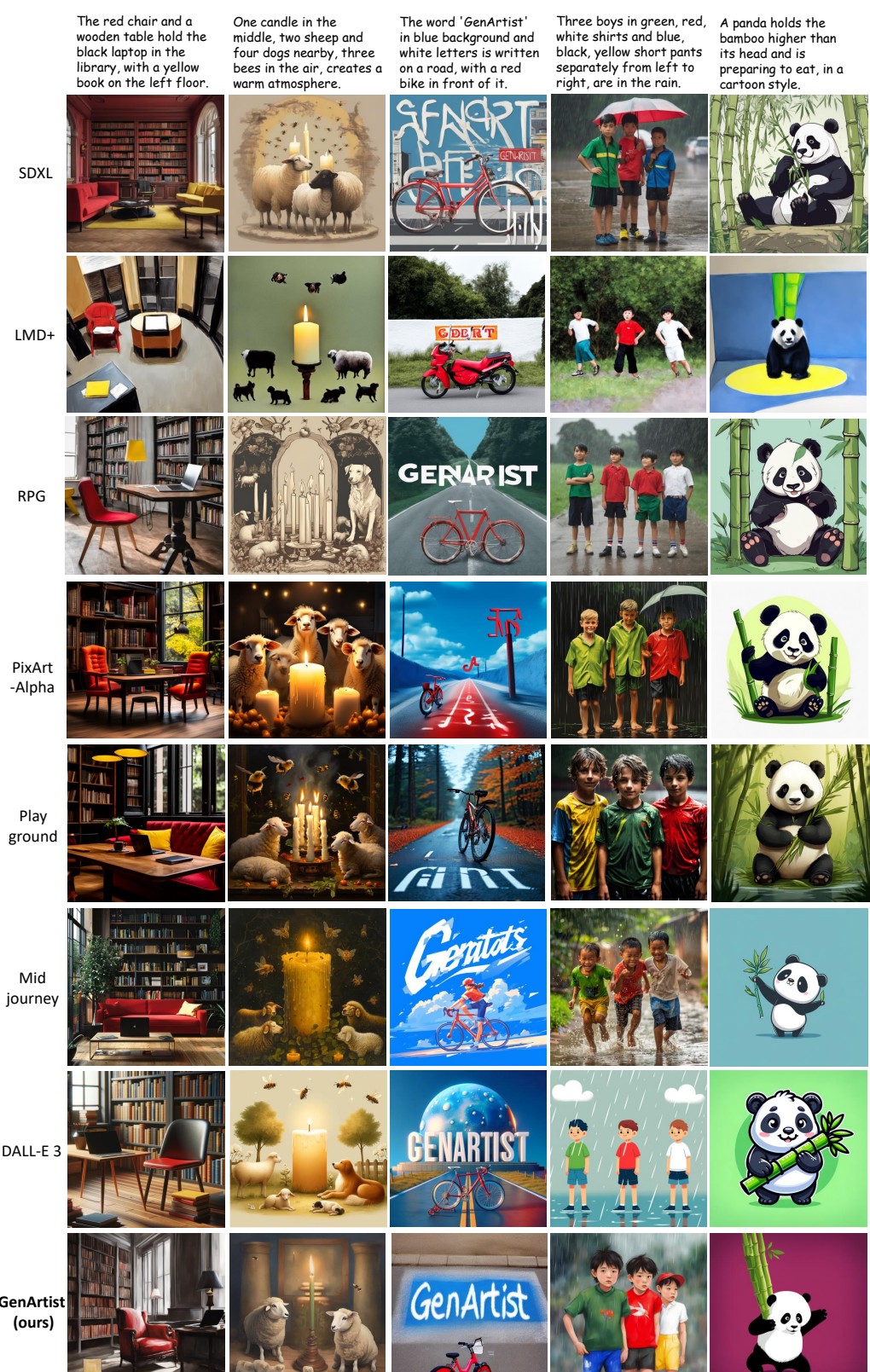

Figure 7: **Qualitative comparison with existing state-of-the-art methods for image generation tasks.** We compare our GenArtist with SOTA text-to-image models including SDXL [37], LMD+ [25], RPG [59], PixArt-$\alpha$ [6], Playground [22], Midjourney [31], DALL-E 3 [1].

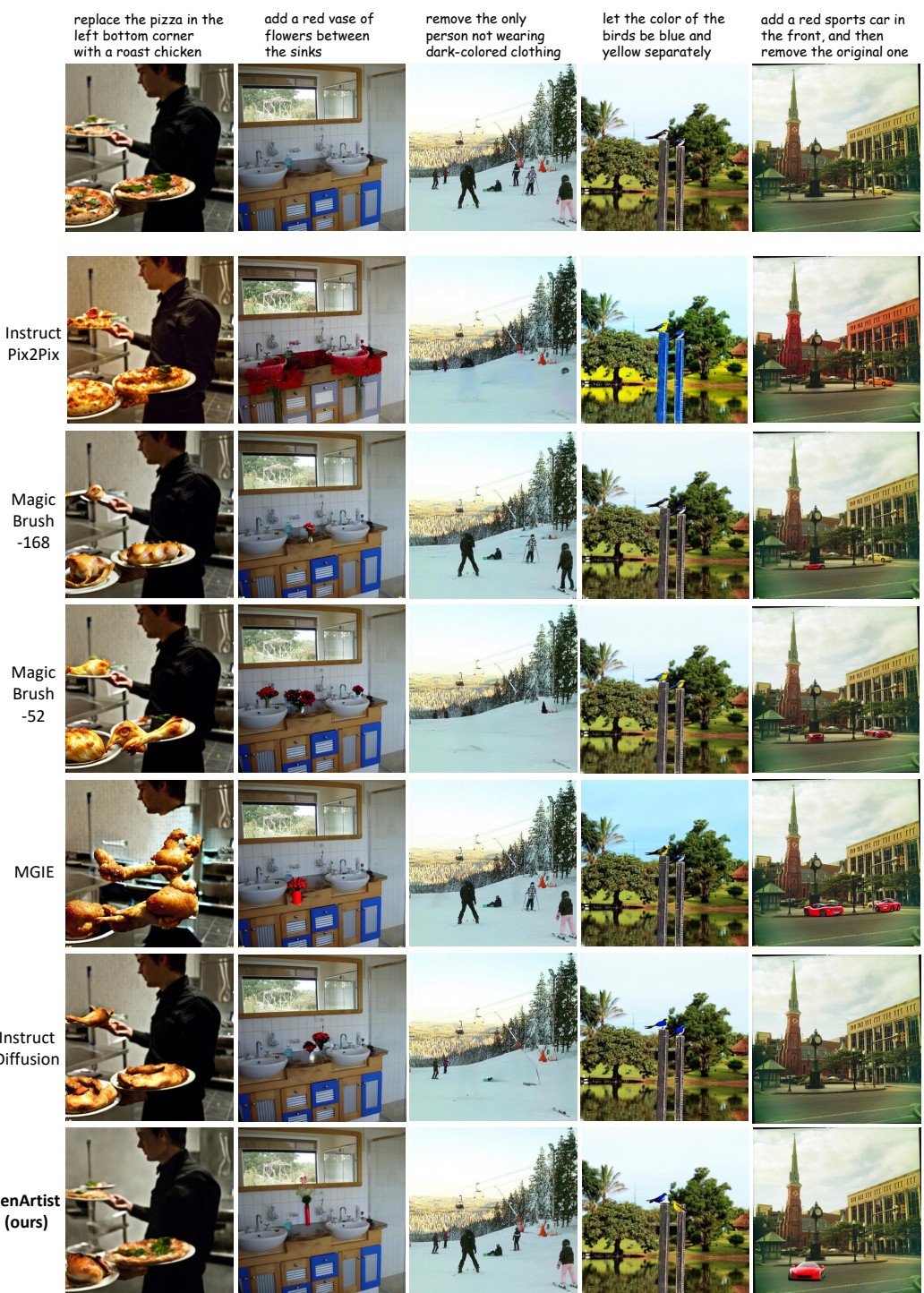

Figure 8: **Qualitative comparison with existing state-of-the-art methods for image editing tasks.** We compare our GenArtist with SOTA image editing models including InstructPix2Pix [2], MagicBrush [61], MGIE [14], InstructDiffusion [15]

third example requires the model to autonomously determine which person needs to be removed. Because of the reasoning capability of the MLLM agent, our method can accurately make this determination. In contrast, even MGIE, which also uses MLLM assistance, fails to make the correct modification. 3) *Instructions with Excessive Requirements*. The fourth example requires different modifications to both birds, which existing methods struggle to achieve. 4) *Multi-step Instructions*.

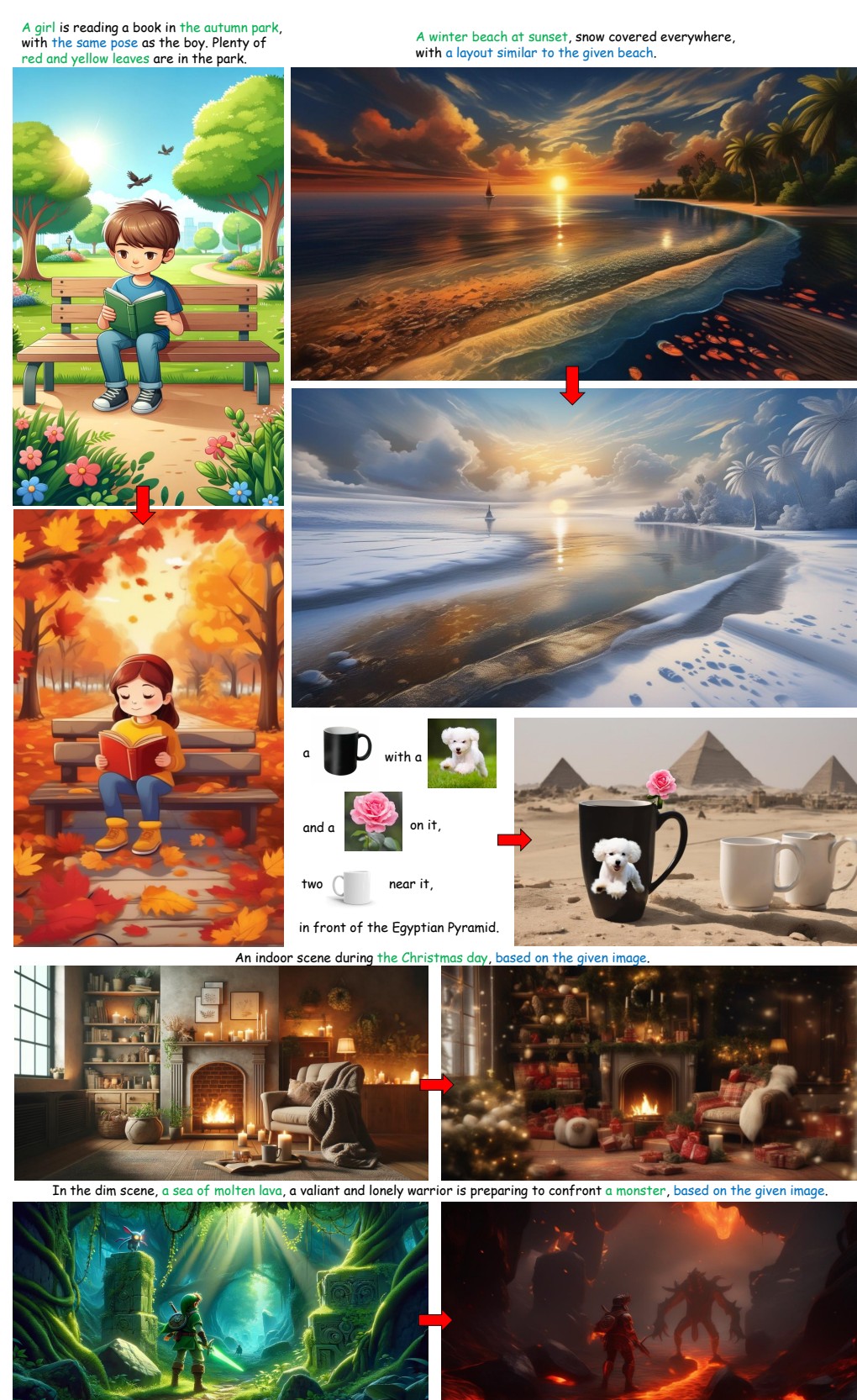

Figure 9: **Visualized results of GenArtist about various tasks and user instructions.**

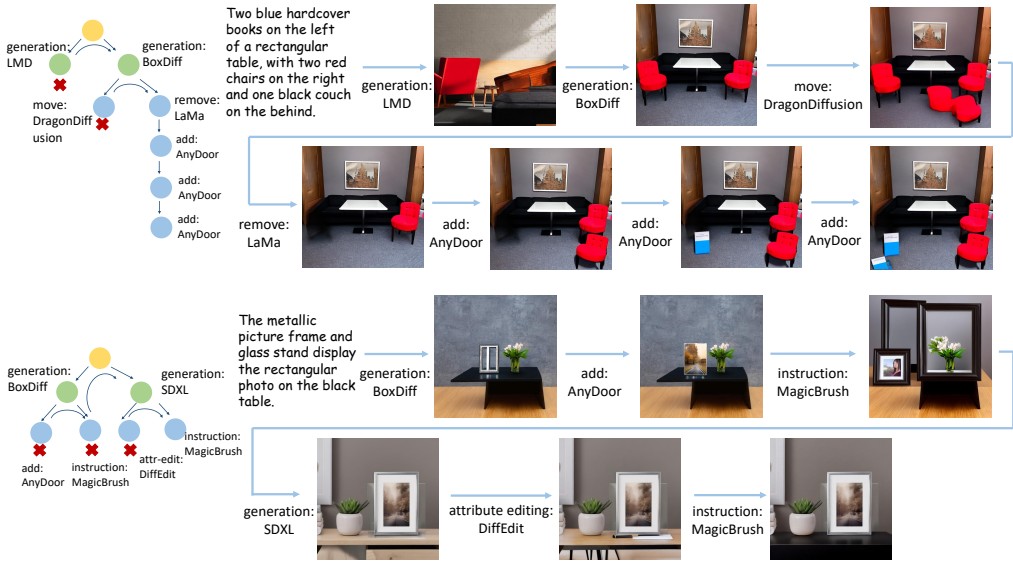

Figure 10: **Visualization of the step-by-step process for image generation tasks.**

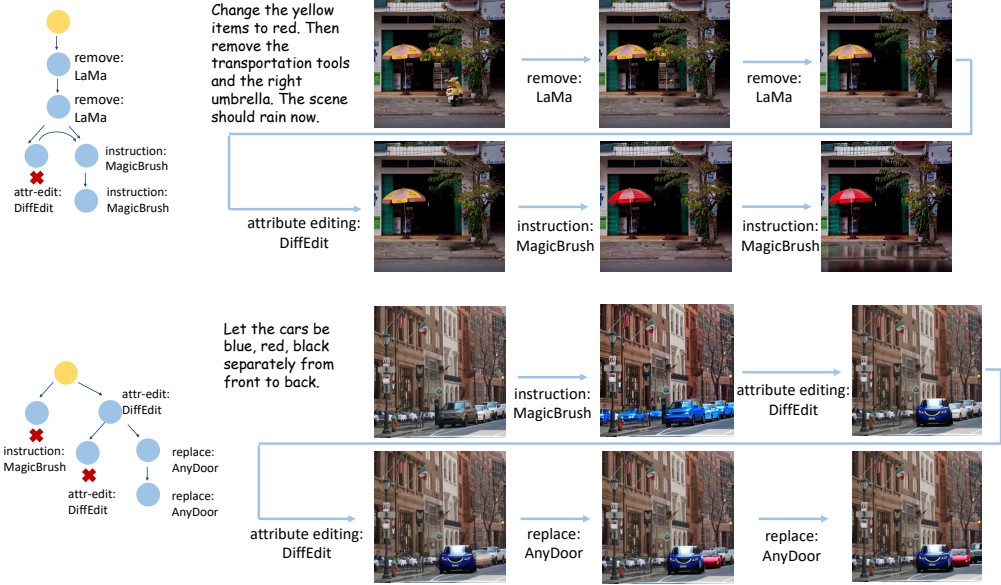

Figure 11: **Visualization of the step-by-step process for image editing tasks.**

The fifth example involves complex instructions including multiple operations. The MLLM agent can decompose the problem into multiple single-step operations, simplifying complex tasks. 5) *Diverse Operations.* It can be seen that our method excels in various editing operations, such as addition, removal, and attribute editing, due to the integration of different tools. These comparisons strongly demonstrate the effectiveness of our method in image editing.

**Visualized results about various tasks and user instructions.** To demonstrate that our GenArtist can meet a wide range of user requirements, we provide visual examples in Fig. 9. As can be seen, because of the integration of various tools, our framework can efficiently address these diverse requirements. For instance, it can generate images with a layout or pose similar to a given image, as well as customization-related generation. Through the use of multiple generation and editing tools, our method also achieves greater control, such as representing more objects and more complex relationships between objects in customization generation. These visualization examples strongly

illustrate the necessity of employing an agent for image generation and demonstrate that our approach effectively accomplishes the goal of unified image generation and editing.

**Visualization for the step-by-step process.** Finally, we present our step-by-step visualized results in Fig. 10 and Fig. 11. For image generation, our method initially utilizes the most suitable tool to generate the initial image. If the image quality is too low or cannot be corrected after some modification operations, additional tools are invoked to continue generation. Further, for parts of the image that do not meet the text requirements, editing tools are continuously called to make modifications until the image correctly matches the text. For image editing, our method effectively decomposes the input problem and iteratively utilizes different tools to make step-by-step modifications until the image is correctly edited. This visualization clearly demonstrates the process, from decomposition and planning tree with step-by-step verification, to the final tool execution.

