# OpenReview forum: "GenArtist: Multimodal LLM as an Agent for Unified Image Generation and Editing"
_NeurIPS.cc/2024/Conference — NeurIPS 2024 spotlight_

### Official Review · Reviewer_tVAw · 2024-07-11

**Soundness:** 3
**Presentation:** 3
**Contribution:** 2
**Rating:** 6
**Confidence:** 4

**Summary:**

This paper presents a method by which an MLLM (gpt 4) is used as a task planner for image editing tasks.  A catalog of off the shelf models is used as a set of tools and the planner constructs a task tree, executes proposed tasks, performs verification, and optionally backtracks.

The authors claim the following contributions:
* a unified image generation and editing system
* a planner which can construct a task tree and perform verification
* the ability to perform tool selection

**Strengths:**

I really like the concept.  It matches intuition that given a strong enough controller / planner, you might be able to perform editing more easily with a bag of tools.

Quantitative evaluations show substantial improvements.

Good writing.

**Weaknesses:**

Probably should cite Gupta, et. al. "Visual Programming: Compositional Visual Reasoning Without Training", CVPR 2023.

No comparison with other MLLMs.

Contributions could be further refined and clarified.

**Questions:**

I would recommend picking one or two more MLLMs and doing a comparison.  I think it's fine that this paper serves as an existence proof that there exists an MLLM with planning capabilities suitable for multi-step image editing, but I didn't get a very good sense of how that performance is impacted by product-specific fine tuning and alignment that might happen behind the scenes with gpt 4.

**Limitations:**

seems fine

---

> ### Author Rebuttal · Authors · 2024-08-05
>
> **Q1: Probably should cite Gupta, et. al. "Visual Programming: Compositional Visual Reasoning Without Training", CVPR 2023.**
>
> A1:
> Thank you for your suggestion. This paper uses language models to generate visual programs for compositional visual tasks, which are then executed. In concept, this paper indeed shares some similarities with AI agent. The difference is mainly that our GenArtist focuses specifically on the field of image generation and editing. In addition to tool execution, we also introduce the verification and self-correction mechanisms to ensure the reliability of the results. In terms of planning, we propose the planning tree method, tailored to the specific characteristics of image generation. We will cite this paper in the related work section.
>
> **Q2: No comparison with other MLLMs.**
>
> A2:
>
> Table 3-1: The peformance of GenArtist on T2I-CompBench, with various MLLMs.
> | |color|shape|texture|
> | :--: | :--: | :--: | :--: |
> |llama2 (70B) + llava|0.7874|0.6223|0.6876|
> |Mixtral-8x7B + llava|0.8142|0.6467|0.7342|
> |GPT4-V|0.8482|0.6948|0.7709|
>
> Since it is relatively difficult for most open-source MLLMs to output in the user-specified format, which makes it inconvenient for the utilization of the system. For better results, we only utilize GPT-4V as the MLLM agent in the original paper.
>
> To experiment with more MLLMs, we utilize LLaVA for multimodal verification, feeding its outputs into large language models such as LLaMA or Mixtral for generating specific planning results. In this way, we conduct experiments on T2I-CompBench with two more MLLM agents and list the results in the above Tab. 3-1. As can be seen, our method can be applied to various MLLMs, demonstrating its flexibility in agent selection. Although the performance with open-source MLLMs is not as high as with GPT4-V, it still shows a significant improvement compared to existing single models. This further demonstrates the effectiveness of GenArtist.
>
> **Q3: Contributions could be further refined and clarified.**
>
> A3:
> Thanks for your suggestion, we will refine and clarify our contributions in the introduction section of the final version.
>
> Overall, we propose a new paradigm for the field of image generation and editing. Instead of utilizing a single model to directly generate the corresponding image, we delegate different functions to different models. An AI agent manages the utilization and sequence of these different models and performs verification and correction on the outputs. By leveraging the strengths of multiple models and the image understanding capability of the MLLM, our GenArtist significantly enhances the reliability of image generation and editing. We hope this innovative approach can provide new insights for future work in this field.
>
> In terms of method design, compared to other existing agent-related methods, our framework is specifically tailored to the characteristics of the image generation and editing field. To address the challenges of complex prompts and the often unreliable generated images, we conduct decomposition and design a planning tree for task planning. Additionally, we introduce position information to enhance the spatial knowledge of the MLLM agent. These designs enable our GenArtist to more effectively tackle image generation related tasks, resulting in improved performance.

---

> > ### Comment · Reviewer_tVAw · 2024-08-12
> > **Rebuttal response**
> >
> > Rebuttal looks good, thank you for the MLLM comparison.  I'm bumping my rating up to weak accept.

---

> > > ### Author Response · Authors · 2024-08-14
> > > **Thanks**
> > >
> > > Dear Reviewer
> > >
> > > We are happy that our reply address your concerns on MLLM comparison. We are sorry to bother you, but It seems that the score has not been changed yet. It will be nice to change the score when you are in convenience.
> > >
> > > Thanks

---

### Official Review · Reviewer_TXUv · 2024-07-12

**Soundness:** 3
**Presentation:** 3
**Contribution:** 2
**Rating:** 5
**Confidence:** 4

**Summary:**

This paper introduces a new multi-modal agent  for image generation and editing that break down these tasks into subproblems to solve with external tools, including self correction module with verification feedback.

**Strengths:**

-	While the idea of augmenting a large (multimodal) language model with tools and turning it into an agent for complex tasks is not new, this work presents a novel multi-modal agent for image generation and editing that combines different existing ideas such as tool use, tree planning, and verification, etc.
-	The authors demonstrate strong positive results on two benchmarks for image generation and editing respectively compared to baselines, which wouldn’t have been possible without great execution.
-	The authors conducted ablation studies and present evidence that shows the importance of the different components e.g. planning tree vs. chain in the system.
-	The paper is well written with informative and clear illustrations.

**Weaknesses:**

-	While the experiments authors conducted are well-motivated and support their claims, the paper would be stronger if it performed a deeper analysis, for example, on common error cases, or via more finegrained ablation studies on the tools / positive-aware tool execution.
-	It would also be stronger if the authors evaluated the system on additional benchmarks such as GenAI-Bench: A Holistic Benchmark for Compositional Text-to-Visual Generation, although the reviewer understands that it might be impossible due to short rebuttal period.
- the paper lacks some technical details about the method such as the underlying model.

**Questions:**

-	Did authors use language-only GPT4 or GPT4-V or GPT4o? For reproducibility, which version of GPT4 was used?
-	One of the common failures of such multi-modal agent systems is due to errors in tool outputs. I wonder if the same issue exists in this system, and if so how the authors address such issues e.g. when the bounding boxes output by the localization/detection module are wrong?

Nits:  table3: why is smart Edit missing L2 number?

**Limitations:**

-	No, the authors have not addressed the limitations of the work.

---

> ### Author Rebuttal · Authors · 2024-08-05
>
> **Q1: The paper would be stronger if it performed a deeper analysis, for example, on common error cases, or via more finegrained ablation studies on the tools / positive-aware tool execution.**
>
> A1:
> * error cases: We show two error cases in the Fig. 2 and Fig. 3 of the rebuttal document. As can be seen, sometimes, despite the agent correctly planning the specific execution of tools, the limitations of the tools themselves prevent correct execution, leading to incorrect results. For example, in Fig. 2 of the rebuttal document, it is required to add a very small blue cup. However, due to the lack of fine resolution ability in existing editing tools, the generated blue cup's size is inaccurate. In addition, as shown in Fig. 3 of the rebuttal document, errors in the output of localization tools can also affect the final result. For instance, when asked to remove the lettuce in the middle of a sandwich, the segmentation model fails to accurately identify the part of the object, leading to the erroneous removal operation.
> * fine-grained ablation studies on the tools/position-aware tool execution: Regarding position-aware tool execution, we conduct the corresponding ablation study and list the results in the below Tab. 2-1. We evaluate the performance on the spatial and complex aspects of T2I-CompBench. As multimodal large models are usually not sensitive to position information, the performance is limited without the inclusion of position information, only a slight improvement over the tool selection results. After introducing position information, which enhances spatial awareness, there is a significant improvement in both the spatial and complex aspects. This validates the reasonability of our design.
>
> Table 2-1: Ablation study on the position-aware tool execution on the spatial and complex aspects of T2I-CompBench.
> |      |  spatial| complex|
> | :--: | :--: | :--: |
> |w/o position-aware tool execution| 0.4577 | 0.4083|
> |w/ position-aware tool execution | 0.5437 | 0.4499 |
>
> **Q2: It would also be stronger if the authors evaluated the system on additional benchmarks such as GenAI-Bench**
>
> A2:
>
> Table 2-2: The performance of GenArtist on the 'basic' prompts of GenAI-Bench .
> | method |attribute|score|spatial|action|part|overall|
> | :--: | :--: | :--: | :--: | :--: | :--: | :--: |
> |SDXL|0.84|0.84|0.82|0.83|0.89|0.83|
> |DeepFloyd-IF|0.83|0.85|0.80|0.82|0.89|0.83|
> |Midjourney v6|0.88|0.87|0.87|0.87|0.91|0.87|
> |DALL-E 3|0.91|0.90|0.92|0.89|0.91|0.90|
> |GenArtist|0.92|0.90|0.93|0.89|0.92|0.91|
>
> Table 2-3: The performance of GenArtist on the 'advanced' prompts of GenAI-Bench .
> | method |count|differ|compare|negate|universal|overall|
> | :--: | :--: | :--: | :--: | :--: | :--: | :--: |
> |SDXL|0.71|0.73|0.69|0.50|0.33|0.63|
> |DeepFloyd-IF|0.74|0.74|0.71|0.53|0.68|0.66|
> |Midjourney v6|0.78|0.78|0.79|0.50|0.76|0.69|
> |DALL-E 3|0.82|0.78|0.82|0.48|0.80|0.70|
> |GenArtist|0.79|0.82|0.79|0.56|0.78|0.74|
>
> We conduct experiments on the GenAI-Bench and report the VQAscores on the basic and advanced prompts separately on the above Tab. 2-2 and Tab. 2-3. As can be seen, GenArtist achieves better performance in almost all aspects of this benchmark, further demonstrating the effectiveness of our approach.
>
> **Q3: the paper lacks some technical details about the method such as the underlying model.**
>
> A3:
> In our framework, we utilize GPT4-V as our MLLM agent. For our tool library, we have listed the utilized tools in the Tab. 1 of our original paper. For auxiliary tools, we utilize Grounding DINO as the object detector, SAM as the object segmentor, ControlNet utilized models for ControlNet-related auxiliary tools, and language-only GPT4 as the layout generator. We will also add more detailed introductions about these tools in the final version.
>
> **Q4:  For reproducibility, which version of GPT4 was used?**
>
> A4:
> We use GPT4-V as our MLLM agent.
>
> **Q5: I wonder if the same issue exists in this system, and if so how the authors address such issues e.g. when the bounding boxes
>  output by the localization/detection module are wrong?**
>
> A5:
> Regarding error cases, we analyze this issue in the error case section of the "A1" section above and include two cases in the rebuttal document. It can be seen that problems within the tool output or the localization module can lead to some erroneous outputs. Utilizing more powerful tools or incorporating some human feedback during the verification stage can effectively address this issue. For example, in the case shown in Fig. 3, using open-vocabulary segmentation models that are capable of recognition or integrating human feedback to adjust the segmentation mask can effectively address this issue.
>
> **Q6: table3: why is smart Edit missing L2 number?**
>
> A6:
> We directly copy the MagicBrush benchmark results from the original SmartEdit paper. Since the original paper only reports four other metrics, we do not include the L2 metric value here.

---

> > ### Author Response · Authors · 2024-08-14
> > **Looking forward to Feedback as Discussion Deadline Approaches**
> >
> > Thanks for your thorough reviews, which are very helpful to improving the quality of our paper. We apologize for any inconvenience caused, but as the deadline for discussion (Aug 13 11:59 pm AoE) draws near, we would like to provide an update on our progress.
> >
> > If you need further clarification or have additional questions, please don't hesitate to contact us. Again, we sincerely thank you for your time and effort in reviewing our paper.
> >
> > Thanks

---

> ### Comment · Area_Chair_HJzT · 2024-08-12
> **Concerns addressed?**
>
> Dear reviewer, thank you for providing constructive feedback to the authors! Are your questions satisfactorily addressed by the rebuttal? Would you like to revise your rating or do you need any more information from the authors to make that decision?

---

> ### Comment · Reviewer_TXUv · 2024-08-14
>
> Thanks for the response and additional evaluations! My questions have been addressed and I have no other concerns. I will keep my rating as it is. Please update the revision with additional technical details e.g. GPT4 version for reproducibility.

---

### Official Review · Reviewer_tJSg · 2024-07-13

**Soundness:** 2
**Presentation:** 4
**Contribution:** 3
**Rating:** 6
**Confidence:** 5

**Summary:**

This paper presents GenArtist, a system that utilizes MLLMs as agents for image generation and editing, especially for complex language descriptions. The key idea is to first use MLLM to decompose the generation task as an execution tree of various tools such as SDXL and LMD, and then utilize all the tools in a predefined tool library. Extensive experiments on T2I-CompBench and MagicBrush shows promising results.

**Strengths:**

- The problem of unified image generation and editing over arbitrary language instructions is very relevant for both academia and industry. How to leverage the existing powerful LLMs such as GPT-4 and other tools is also very important for the deployment of current academic progress.
- The proposed method is quite simple and is reasonable. The paper is clearly written and the results look good to me.

**Weaknesses:**

Overall, I think this paper has made some good contributions and shown some promising results. However, I hold some concerns as follows,

**Major concern.** My major concern is about the possible limitation in the decomposition tree. All images generated for the complex text is by first generating an initial image, and then utilizing editing tools to match the text description. This could be very limited if the images initially generated are not very decent. This is because even if the final edited image matches the language description better in semantics, the overall quality and layout may not be very natural in some cases. The editing trace may be easily detected and this may be attributed the weakness of the editing models. The editing operation in the paper is mostly editing local regions while keeping the original overall structure, and this could be the problem. For example, the generated result with "hot dogs" in Figure 1 is not natural or of decent quality to me when compared to results generated by models like Playground. The image is blurred and the hot dogs look a little fake.

**Human-aligned evaluation.** The current evaluation is mainly conducted on T2I-CompBench and MagicBrush. However, it has been well known that traditional metrics such as DINO scores are not aligned with humans. This could cause a result that even if the image quality is not as good as another single-expert result, the score may still be better. However, this might be misaligned with human preference in some cases. Can authors conduct more evaluations on more advanced benchmarks like DreamBench++ [1]?

Minor suggestion: I think the paper organization could be improved by trying to put each full paragraph on one page or two. Try your best not to split one section or subsection into two pages. For example, Section 4.1 and 4.3 are all put under the bottom of one page with very few contents, this could cause a sense of reading fragmentation.

[1] DreamBench++: A Human-Aligned Benchmark for Personalized Image Generation.

**Questions:**

As stated in weakness, I am curious about the results of modern human-aligned benchmarks, such as DreamBench++. I would be happy to see more human-aligned evaluation, if possible.

I am looking forward to the authors' response.

**Limitations:**

Yes, the authors have discussed the limitations. However, I would suggest the authors discuss the potential limitation brought by the initial generated results and editing artifacts.

---

> ### Author Rebuttal · Authors · 2024-08-05
>
> **Q1: My major concern is about the possible limitation in the decomposition tree.**
>
> A1:
> * Thank you for your suggestion. We have indeed considered this issue. Therefore, during verification, in addition to verifying the accuracy of the generated images, the agent is also required to assess their aesthetic quality (as illustrated in L173). If the overall quality of the generated image is poor, the agent will utilize different generation tools or choose different random seeds to regenerate the images, in order to ensure their overall quality. For instance, in Fig. 9 of our main paper, the agent regenerates the image due to the low quality of the initially generated image, thereby ensuring the aesthetic quality of the final generated image.
> * By explicitly requiring the agent to maintain higher standards for the aesthetic quality of generated images, the overall quality of the generated images can be further improved. We apply this to regenerate the "hot dogs" case in Fig. 1, and list the generated image in the Fig. 1 of the rebuttal document. As can be seen, the image quality is further improved.
> * The quality issue is indeed related to the utilized editing tools. The main reason is that most of the currently utilized editing tools are based on Stable Diffusion 2 or even v1.4. Due to the weaker based models, these editing tools are usually limited in keeping the overall image quality compared to larger generative models like SDXL. We can expect that as more powerful editing models emerge, we can replace the corresponding models in the current editing tool library and this issue will be alleviated.
> * As an agent-centered system, our framework is also flexible in terms of human-computer interaction. During verification, human feedback can be appropriately integrated. By incorporating human evaluation and feedback on the overall quality of the images, the quality of the generated images can be further improved.
>
> **Q2: Can authors conduct more evaluations on more advanced benchmarks like DreamBench++?**
>
> A2:
>
> Table 1-1: The performance on the DreamBench++ benchmark in the GPT score.
> |   method   | concept preservation  | prompt following |
> | :--: | :--: | :--: |
> |BLIP-Diffusion | 0.547 | 0.495|
> |Emu2 | 0.528 | 0.689 |
> |IP-Adapter-Plus | 0.833 | 0.413 |
> |GenArtist| 0.848 | 0.603 |
> |GenArtist (more tools) | 0.852 | 0.753 |
>
>
> We conduct experiments on DreamBench++ and compare our results (in GPT score) with several tuning-free methods listed in the original DreamBench++ paper. Since DreamBench++ is primarily about single-object customization generation, where our current GenArtist framework includes only a few relevant tools, we also expand our tool library by introducing more customization tools, and then conduct the corresponding experiments. The comparative results are listed in the above Tab. 1-1.
>
> As can be seen, the current version of GenArtist has already achieved the strong performance on DreamBench++, surpassing all listed tuning-free methods in concept preservation and outperforming most existing models in prompt following. After expanding the tool library, further improvement can be observed, particularly in prompt following. This demonstrates that our framework can achieve better results no matter for image accuracy or image quality on such a recent benchmark.
>
> **Q3: I think the paper organization could be improved by trying to put each full paragraph on one page or two.**
>
> A3:
> Thank you for your suggestion. We will enhance the paper organization by adding some additional explanations, such as introducing aspects about image quality, to ensure that sections 4.1 and 4.3 do not span across pages.

---

> ### Comment · Reviewer_tJSg · 2024-08-11
> **Thanks, rating kept**
>
> Thank you for your rebuttal efforts and response. The hot dog case does look better but it is still a little bit fake. However, this issue can be alleviated by using more advanced image editions and generative tools. I appreciate you for letting me know about the aesthetic quality correction procedure, which is helpful. I strongly suggest that the authors discuss this issue properly in the revised paper and incorporate all rebuttal experiments into the final paper. Besides, I suggest the authors use the latest GPT-4o instead of GPT-4V and conduct all experiments again since GPT-4o is a more advanced GPT tool to date, and I think it would be valuable to keep using state-of-the-art tools. All details, including the precise GPT-4o version (eg, `turbo-2024-04-09`), should be explicitly recorded in the paper.
>
> Overall, I think my concern is resolved, and I will keep my rating for a supportive assessment for this paper.

---

### Author Rebuttal · Authors · 2024-08-05

We include some essential images for the rebuttal in the PDF file here, mainly comprising the regenerated images for the hot dog case and some analysis about error cases.

---

### Decision · Program_Chairs · 2024-09-25

**Decision:**

Accept (spotlight)

**Comment:**

The paper proposes a system for image editing and generation that uses a MLLM for execution-tree based planning with verification and existing tools for execution. The work received unanimously positive reviews, particularly with the reviewers acknowledging their concerns were addressed by the rebuttal.

The reviewers appreciated the novel and timely idea, good results and ablations, clear writing, and the incorporation of planners and verifies for tool-augmented image editing/generation.

The AC agrees with the reviewers and therefore recommends accepting. Congratulations to the authors!